

# Starch biotransformation into isomaltooligosaccharides using thermostable alpha-glucosidase from *Geobacillus stearothermophilus*

Peng Chen[1,2], Ruixiang Xu[1], Jianhui Wang[2], Zhengrong Wu[1], Lei Yan[3], Wenbin Zhao[1], Yuheng Liu[1], Wantong Ma[1], Xiaofeng Shi[4] and Hongyu Li[1]

[1] School of Pharmacy, Lanzhou University, Lanzhou, Gansu, PR China
[2] School of Medicine, Yale University, New Haven, CT, United States of America
[3] College of Life Science and Technology, Heilongjiang August First Land Reclamation University, Heilongjiang, PR China
[4] Gansu Academy of Medical Science, Lanzhou, PR China

Corresponding author
Hongyu Li, lihy@lzu.edu.cn

## ABSTRACT

The present study first identified the biotransformation of starch as a novel preparation method was investigated using the alpha-transglucosidase-producing *Geobacillus stearothermophilus* U2. Subsequently, 5 L- and 20 L-scale fermentations were performed. After isolation and purification, liquid alpha-glucosidase preparations were obtained. Through covalent cross-linking and adsorption cross-linking using chitosan as the carrier and glutaraldehyde as the crosslinking agent, the conditions for immobilization of alpha-glucosidase on chitosan were determined. Moreover, Isomaltooligosaccharides (IMOs) were then prepared using chitosan membrane-immobilized alpha-glucosidase, beta-amylase, pullulanase, fungal alpha-amylase and starch as substrate. The mixed syrup that contained IMOs was evaluated and analyzed by thin-layer chromatography (TLC) and high-performance liquid chromatography (HPLC). In addition, small-scale preparation of IMOs was performed. These results are a strong indication that the alpha-transglucosidase-producing *G. stearothermophilus* as a potential application technique can be successfully used to prepare industrial IMOs.

## INTRODUCTION

Alpha-glucosidases (EC3.2.1.20) belong to the starch hydrolase family and mainly exert their functions outside of cells (*Hirschhorn, Huie & Kasper, 2002*). Alpha-glucosidase hydrolyzes the alpha-glycosidic bond from the non-reducing end of the polysaccharide substrate, releasing alpha-D-glucose. Alpha-glucosidases are generally categorized as hydrolytic enzymes (class 3) and mainly hydrolyze disaccharides, oligosaccharides, aromatic glycosides, sucrose and polysaccharides (*Mohamed Sham Shihabudeen, Hansi Priscilla & Thirumurugan, 2011*; *Verastegui-Omaña et al., 2017*). In addition, alpha-glucosidase mediates transglycosidation (*Johnson et al., 2016*). The enzyme converts the

alpha-1,4-glucosidic bond in polysaccharides into an alpha-1,6-glycosidic bond or other forms of linkage (*Benayad et al., 2016*), resulting in the formation of non-fermentable isomaltooligosaccharides (IMOs) or sugar esters/glycopeptides (*Kaneko et al., 1995*). In the 1980s, Japanese scientists first isolated alpha-glucosidase-producing fungal strains of *Aspergillus niger*. Since then, alpha-glucosidase has been widely used. In the starch and sugar industry, alpha-glucosidase is mainly used, together with alpha-amylase, in the production of high-glucose syrup.

Alpha-glucosidase is also used to produce IMOs that function as bifidus factors (*Lambert & Zilliken, 1965*). IMOs, also known as branched oligosaccharides, are a type of syrup produced by full enzymatic processing using starch as the raw material (*Kaneko et al., 1995*). IMOs consist of 2–10 glucose residues and contain at least one alpha-1,6-glycosidic bond and the remaining bonds in IMO molecules are all alpha-1,4-glycosidic bonds. Commercial IMOs are a class of syrups that contain isomaltose, panose, isomaltotriose and branched oligosaccharides composed of more than four sugar residues (*Chockchaisawasdee & Poosaran, 2013*). The branched oligosaccharides account for more than 50–55% of the total sugar content. Other sugars in commercial IMOs include glucose and maltose. The sweetness of commercial IMOs is 40–50% of the sweetness of sucrose. IMOs promote the growth of probiotics such as *Bifidobacterium* and *Lactobacillus*, adjust the balance of intestinal flora, promote peristalsis, prevent and relieve constipation and diarrhea, prevent dental caries, and inhibit the growth of harmful intestinal bacteria and the formation of decaying substances. Through the proliferation of *Bifidobacterium* and other probiotics, IMOs indirectly exert many health protection functions, including anti-tumor function, liver protection, promotion of vitamins, improved immunity, and prevention of cardiovascular and cerebrovascular diseases (*Maina et al., 2011*). However, most of the alpha-glucosidases reported in the literature cannot be used for the industrial production of oligosaccharides because most alpha-glucosidases are not thermostable. Therefore, to meet different production requirements, the development of novel, high temperature-resistant alpha-glucosidases with good stability is necessary (*Khang, Phuong & Ma, 2017*).

Alpha-glucosidases are widely distributed in nature, and there are a great variety of alpha-glucosidases (*Xiao et al., 2012*). These alpha-glucosidases display distinct characteristics and exist in almost all organisms. Among the alpha-glucosidases that have been studied, the vast majority are derived from microorganisms while only a few are from plants and animals. *Aspergillus niger* (*Liu et al., 2013*; *Hatano et al., 2017*), *Aspergills foetidus*, *Bacillus licheniformis* (*Nawaz et al., 2014*; *Nawaz et al., 2016*), *Pseudoalteromonas* sp. (*Li et al., 2016*), *Podospora anserina* (*Song et al., 2013*), *Monascus rubervan Tieghem* (*Chen & Xie, 2008*), crenarchaeon *Sulfolobus tokodaii* (*Park et al., 2013*), *Patinopecten yessoensis* (*Masuda et al., 2016*), *Geobacillus* sp (*Hung et al., 2005*), *Malbranchea cinnamomea* (*Yan et al., 2015*), *Saccharomyces cerevisiae* (*Hossain et al., 2016*), *Xanthophyllomyces dendrorhous* (*Gutierrez-Alonso et al., 2016*), *Thermus thermophilus* (*Zhou, Xue & Ma, 2015*), rice seed (*Kim et al., 2016*), *Bifidobacterium adolescentis*, *Geobacillus stearothermophilus*, *Bombyx mori*, and *Spodoptera frugiperda* (*Watanabe et al., 2013*) secrete alpha-glucosidase. Among the microorganisms mentioned above, *A. niger* produces a larger amount of alpha-glucosidase (*Giles-Rivas et al., 2016*). Most of the commercially available alpha-glucosidase products

are synthesized through *A. niger*-mediated fermentation, such as the transglucosidase from Amano Enzyme Inc., Japan. *G. stearothermophilus* U2 belongs to the thermophilic aerobic bacilli and is a Gram-positive bacterium. Enzymes produced by *G. stearothermophilus* U2 possess heat-resistant properties. Application of immobilization technology not only enhances enzymatic activity but also achieves better reusability of the enzymes. The fundamental knowledge derived from this study should provide a valuable platform for further investigation into the behavior of *G. stearothermophilus* involved in starch biotransformation and has potential biotechnological applications in IMOs production.

Therefore, the behavior of alpha-transglucosidase-producing *G. stearothermophilus* U2 in response to different fermentation conditions were investigated in a series of batch experiments. After isolation and purification, liquid alpha-glucosidase preparations were obtained. Through covalent cross-linking and adsorption cross-linking using chitosan as the carrier and glutaraldehyde as the crosslinking agent, the conditions for immobilization of alpha-glucosidase on chitosan were determined. Starch was liquefied under the action of thermostable alpha-amylase. Liquefied starch was converted into maltose syrup by mesophilic alpha-amylase or bata-amylase. Immobilized alpha-transglucosidase-mediated transglycosylation was then conducted to produce IMOs. The mixed syrup that contained IMOs was tested and analyzed by thin-layer chromatography (TLC) and high-performance liquid chromatography (HPLC).

## MATERIALS AND METHODS

### Organism

A culture of *G. stearothermophilus* U2 was purchased from the China General Microbiological Culture Collection Center (CGMCC). These bacteria were cultured in nutrient broth medium and used throughout the experiments. The alpha-transglucosidase-producing *G. stearothermophilus* U2 was grown aerobically at 60 °C for 17 h in a medium containing 2% (w/v) soluble starch, 2% peptone, 0.05% meat extract, 0.2% yeast extract, 0.3% $K_2HPO_4$ and 0.1% $KH_2PO_4$ (pH 7.0), as described previously (*Suzuki, Shinji & Eto, 1984*). The corn starch used in the present study was food-grade and commercially available. All other reagents were analytical grade.

### Enzyme assay

Alpha-transglucosidase activity was assayed using p-NPG (P-nitrophenol-alpha-D-glucoside) as substrate (*Suzuki, Shinji & Eto, 1984*). p-NPG was purchased from Sigma-Aldrich Corporation. The standard reaction system 1.0 ml consisted of 33.3 mM phosphate (pH 6.8), 2 mM p-NPG and enzyme. Incubation was for 20 min at 60 °C. One unit of the activity was defined as the amount of enzyme hydrolyzing 1 μmol of the nitrophenyl glucoside/min under the condition described above (*Suzuki, Shinji & Eto, 1984*).

### Effects of environmental parameters on enzyme production

Figure 1 shows that the flow-process diagram of starch biotransformation into isomaltooligosaccharides using thermostable alpha-glucosidase from *G. stearothermophilus*. All batch experiments were carried out with microorganism suspension in 250-ml shake

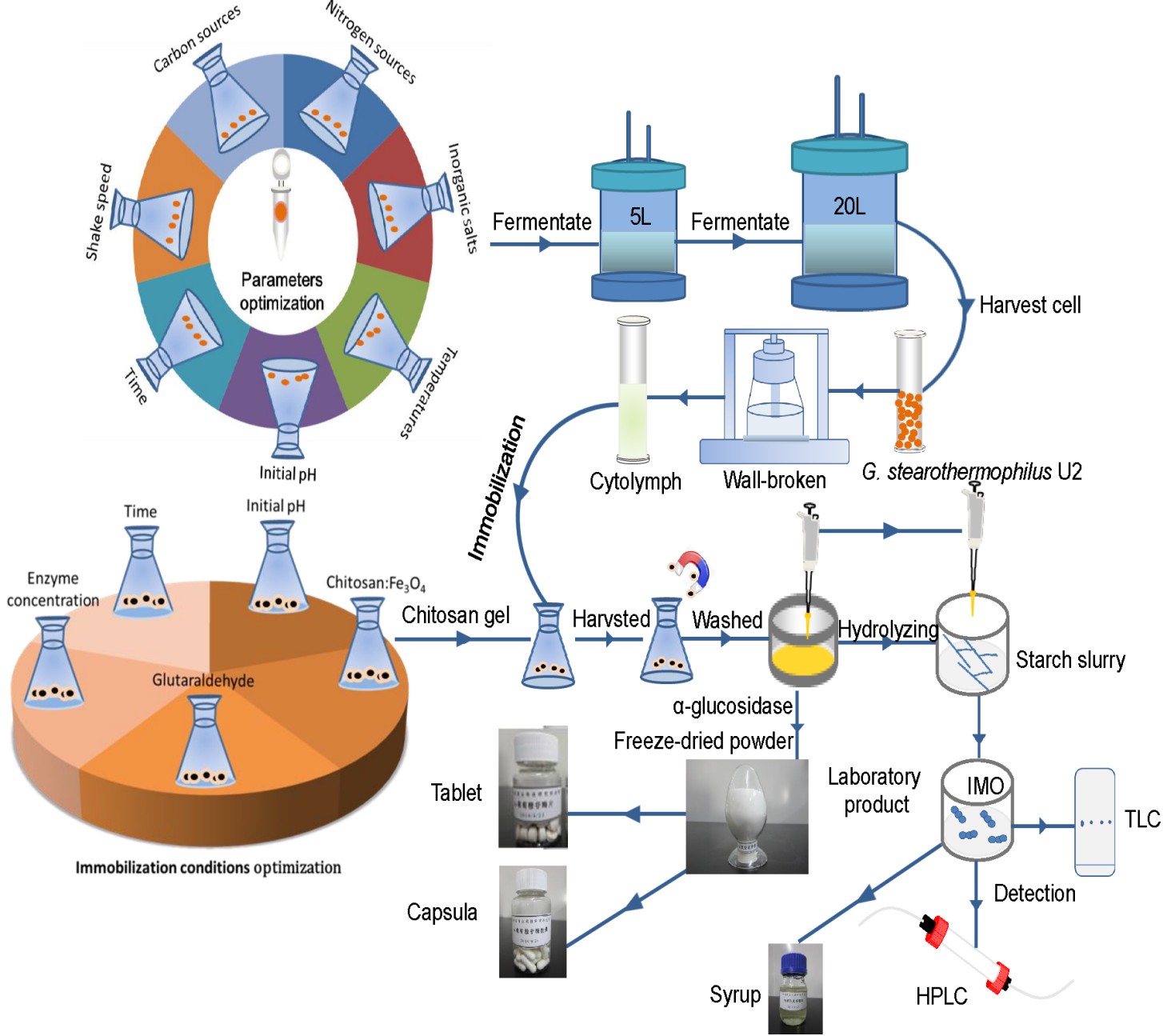

**Figure 1** The flow-process diagram of starch biotransformation into isomaltooligosaccharides using thermostable alpha-glucosidase from *G. stearothermophilus.* Source credit: Peng Chen.

flask at 150 r/min. A series of culture medium were prepared by adding different carbon sources (glucose, starch, maltose, lactose or sucrose; 0.5 g/L), nitrogen sources (beef extract, peptone, yeast powder or ammonium sulfate; 0.5 g/L) and inorganic salts ($MgCl_2$, $K_2HPO_4$, $CaCl_2$, $MnCl_2$, $CoCl_2$, $FeCl_2$, $ZnCl_2$ and $CuCl_2$) to basic potato medium. After culturing the bacterial strain in different media for 24 h, at pH 7.0 and 37 °C with shaking

(200 r/min), enzyme activity was measured. In addition, enzyme activity was examined at different temperatures (30, 33, 37, 40 and 42 °C), various time points (12, 16, 20, 24, 28 and 32 h of culture), different initial pH (6.2, 6.5, 6.8, 7.1 and 7.4) and different shaker rotational speeds (150, 200, 250 and 300 r/min). A sterile control flask without bacteria and under identical experimental conditions ran concurrently. Fermentation experiments were carried out in a 5 L fermenter (B.E. Marubishi, Japan) and a 20 L fermenter (Zhenjiang East Biotech, PR China) based on the test tube and shaker flask cultures. We evaluated the effects of a range of crucial operational parameters, including inoculation volume, 10%; ventilation rate, 5 L/min; rotational speed, 500 r/min; initial temperature, 37 °C; initial pH, 6.8–7.2; and initial dissolved oxygen (DO) value, 99. After fermentation for 8 h, the DO value was reduced to zero. Once the level of bubbles reached the alarm value, fermentation was terminated. The experiments lasted approximately 20 h. All experiments were conducted in triplicate and the average values were reported.

## Immobilization of alpha-glucosidase on magnetic chitosan microspheres

The preparation method for immobilization alpha-glucosidase on magnetic chitosan microspheres was previously described (*Denkbaş et al., 2002*). Firstly, preparation of magnetic chitosan microspheres: Nano $Fe_3O_4$ particles and chitosan were added to an acetic acid solution at a specific ratio. After 24 h of incubation, the mixture was subjected to ultrasonic dispersion for 30 min. Subsequently, 70 mL of liquid paraffin and 7 mL of Span-80 were added to a triangular flask and mixed thoroughly by stirring at 40 °C for 20 min. The $Fe_3O_4$ particle-containing chitosan solution was then added dropwise to the flask. After agitation for 30 min, 5 mL of 4% glutaraldehyde solution was added and allowed to react for 1 h. Afterwards, the pH of the reaction mixture was adjusted to approximately 10.0 with 1 mol/L NaOH solution. The mixture was then heated to 60 °C and allowed to react for another 2 h. To recover the magnetic chitosan microspheres, the reaction products were collected using a magnet, washed successively with petroleum ether, acetone and distilled water, and vacuum-dried at 60 °C (*Denkbaş et al., 2002*). Secondly, immobilization of alpha-glucosidase on magnetic chitosan microspheres. Eight micrograms of magnetic chitosan microspheres were immersed in 2 mL of 1.5% acetic acid solution for 24 h, allowing for the full swelling of the microspheres. Six microliters of 0.1 mol/L NaOH solution was then added to the microspheres. Alpha-glucosidase was dissolved in 0.1 mol/L phosphate buffer (pH 6.8) to prepare a solution with an enzyme concentration of 6.5 mg/mL. The prepared enzyme solution was then added to the chitosan gel. Enzyme adsorption was achieved by oscillation at 30 °C and 150 r/min for 1 h. Subsequently, crosslinking of the enzyme was conducted by addition of 2 mL of 1% glutaraldehyde solution and oscillation at a constant speed of 150 r/min for 1 h at 60 °C. The chitosan gel was then washed with 0.1 mol/L phosphate buffer (pH 6.8) until no free enzyme could be detected by the ninhydrin method. The immobilized alpha-glucosidase was collected using a magnet, and the activity of the immobilized enzyme was determined. Lastly, the activity recovery rate of the immobilized enzyme was calculated as follows: activity recovery rate of the immobilized enzyme (%) = $X_1/X_2$, where $X_1$ was the total activity of the immobilized

enzyme (m mol/g) and $X_2$ was the total activity of the added free enzyme (m mol/g). Protein detection by the ninhydrin colorimetric method has been described previously (*Hung et al., 2005*).

## The effects of immobilization conditions on the efficiency of alpha-glucosidase immobilization

Chitosan and $Fe_3O_4$ were added to a 1% acetic acid solution at ratios of 1:0.5, 1:1, 1:2, 1:3 and 1:4 successively. The mixtures could sit for 24 h and then subjected to ultrasonic dispersion for 30 min. The pH of the resulting chitosan gel systems was adjusted to approximately 10.0 with 1 mol/L NaOH, and the states of the systems were examined. Subsequently, 0.5, 1.0, 1.5, 2.0, 2.5, 3.0 and 4.0 mL of enzyme solution (enzyme concentration: 6.5 mg/mL) were added to the chitosan gel solutions successively, and the activity of the immobilized enzyme was determined. Alpha-glucosidase was then added to the chitosan gel solutions at a certain concentration and allowed to absorb for 0.5, 1, 1.5, 2, and 2.5 h. The effects of various concentrations of glutaraldehyde (1, 2, 3, 4 and 5%) on enzyme immobilization were determined. Crosslinking of the adsorbed enzyme was then carried out for a certain length of time at 35, 40, 45, 50 and 55 °C in the presence of 4% glutaraldehyde. In addition, the crosslinking duration was varied (0.5, 1.0, 1.5, 2.0 and 2.5 h). The activity of the immobilized enzyme was measured under each condition.

## Optimization of the transglycosidation process

An appropriate dextrose equivalent (DE) value was used as an indicator (in the present study, a DE value of 12 was used). The DE value was determined using the iodometric method (*Lin et al., 2012*). The effects of various factors on the process of liquefaction were individually investigated. Corn starch was weighed, and the desired amounts of corn starch were added to distilled water, creating aqueous slurries with 15%, 20%, 25%, 30% and 35% starch. The starch slurries were preheated in a 95 °C water bath for 5 min. Subsequently, thermostable alpha-amylase (15 U/g) was added to the starch slurries. After a 10-min liquefaction, the alpha-amylase was inactivated by increasing the acidity and raising the temperature. The effect of substrate (starch) concentration on the DE value of the liquefied solution was examined. The effect of the dosage of thermostable alpha-amylase on the DE value of the liquefied solution was determined using the following procedure: the concentration of the starch slurry was adjusted to the optimum based on the result of (1). The starch slurries were then preheated in a water bath at 95 °C for 5 min. Subsequently, various doses of thermostable alpha-amylase (10 U/g, 15 U/g, 20 U/g, 25 U/g and 30 U/g) were added to the starch slurries. After a 10-min liquefaction, the alpha-amylase was inactivated by increasing the acidity and raising the temperature. The effect of the dosage of thermostable alpha-amylase on the DE value of the liquefied solution was examined. The effect of liquefaction time on the DE value of the liquefied solution was determined using the following method: The concentration of the starch slurry was adjusted to the optimum based on the result of (1). The starch slurries were then preheated for 5 min in a 95 °C water bath. Subsequently, the optimal dose of thermostable alpha-amylase, which was determined in (2), was added to the starch slurries.

**Table 1** Factors and levels in orthogonal analysis of the liquefaction process.

| Level | Factor | | |
|---|---|---|---|
| | A<br>Starch slurry concentration (%) | B<br>Dosage of liquefying enzyme (U/g) | C<br>Liquefaction time (min) |
| 1 | 1 | 1 | 1 |
| 2 | 2 | 2 | 2 |
| 3 | 3 | 3 | 3 |

After liquefaction for various lengths of time (8 min, 10 min, 12 min, 14 min, 16 min, 18 min and 20 min), the alpha-amylase was inactivated by increasing the acidity and raising the temperature. The effect of liquefaction time was then examined. Based on the results of the single-factor experiments, the experimental factors and levels of the liquefaction process were determined through orthogonal analysis (Table 1). The main factors affecting the saccharification and transglucosidation process included the following: glucoamylase dosage, saccharification time, dosage of alpha-transglucosidase (purchased from Amano Enzyme Inc., Japan, or made in our laboratory) and duration of the transglucosidation process. The isomaltose content was employed as an indicator and was determined by paper chromatography (Zhang et al., 2011). The effects of various factors on the saccharification and transglucosidation processes were determined individually.

## Preparation of IMO

IMO preparation was performed according to the reported method (Goulas et al., 2004). (1) The liquefaction conditions were as follows: corn starch slurry concentration, 20%; dosage of thermostable alpha-amylase, 15 U/g; and liquefaction time, 10 min. (2) The saccharification and transglucosidation conditions were as follows: glucoamylase dosage, 250 U/g; saccharification time, 4.5 h; dosage of transglucosidase, 1 U/g; and the duration of the transglucosidation process, 42 h.

## Analytical procedures

Determination of biomass: One milliliter of the fermentation broth was collected and properly diluted. The absorbance of the fermentation broth at A600 nm was determined, which represented the bacterial density (Suzuki, Shinji & Eto, 1984). The enzymatic activity of alpha-glucosidase was measured using a method described previously (Wu et al., 2010). The method employed to determine the DE value (the amount of reducing sugars) has been described previously.

$$\mathrm{DE(G,\%)} = \frac{c(V_1 - V_2) \times 0.09}{W \times \frac{10}{200}} \times 100$$

where $c$ is the molar concentration of the sodium thiosulfate standard solution (mol/L), $V_1$ is the volume of the sodium thiosulfate standard solution consumed by the blank (mL), $V_2$ is the volume of the sodium thiosulfate standard solution consumed by the sample (mL), $W$ is the mass of fermented sugar (g), and 0.09 represents the number of grams

of glucose that was equivalent to 1 mL of 1 mol/L iodine standard solution. Qualitative analysis of the isomaltose content via paper chromatography was performed using the following developing agent: n-butanol: pyridine: water = 6:4:3. The chromogenic reagent (aniline-diphenylamine-phosphoric acid) was prepared by dissolving 4 g of diphenylamine, 4 mL of aniline and 20 mL of 85% phosphoric acid in 200 mL of acetone. The prepared IMO syrup was spotted onto chromatography paper and exposed to the developing agent for 4 h. The chromatography paper was removed, dried and then exposed to the developing agent again. This step was repeated three times (approximately a total of 12 h of development). Subsequently, the chromatography paper was naturally dried, sprayed with chromogenic agent and heated at 70–80 °C for 10 min. The IMO content was preliminarily determined based on the size (diameter) of the developed spots. A high-performance liquid chromatographic (HPLC) method for IMO measurements was based on reported methods (*Vinogradov & Bock, 1998*). A Waters Alliance 2695 liquid chromatography unit equipped with a 1,515 pump, 717 plus autosampler, a reversed-phase C18 column (250 × 4.6 mm, 5 μm) and a Waters 2996 UV detector was used (Waters Co., Milford, CT, USA). The mobile phase was prepared by adding 270 mL of purified water to 730 mL of acetonitrile, and then filtered under vacuum through a 0.45-μm filter. The HPLC was performed at a flow rate of 1.3 mL/min and an injection volume of 20 μL.

## RESULTS AND DISCUSSION

### The effect of environmental parameters on enzyme production by *G. stearothermophilus* U2

The effects of different carbon sources on the production of enzyme were investigated using *G. stearothermophilus* U2. Among the carbohydrates tested, the highest production of a-glucosidase was achieved by glucose as the carbon source (Fig. 2A). Furthermore, both starch and maltose were capable of serving as effective carbon sources and promoting the synthesis of alpha-glucosidase. In addition, the ability of the enzyme to act on starch and maltose was induced when the two types of sugar were present. In contrast, lactose and sucrose were not conducive to enzyme production. The effects of different nitrogen sources on enzyme production by *G. stearothermophilus* U2 were examined (Fig. 2B). Various nitrogen sources (0.5 g/L) were added to basic potato medium (20% potato, Control). Bacterial strain U2 was then inoculated into the medium. After culture of the bacteria for 24 h, at pH 7.0 and 37 °C with shaking (shaker rotational speed, 200 r/min), enzyme activity was examined. When beef extract, peptone and yeast powder were selected as the nitrogen source, the synthesis of alpha-glucosidase was promoted. Therefore, beef extract, peptone and yeast powder are ideal nitrogen sources for the fermentation medium. The effects of various inorganic salts on enzyme production by *G. stearothermophilus* U2 were also examined. Different inorganic salts were added to the measurement system to achieve a final concentration of 10 mmol/L. Enzyme activity in the inorganic salt-free system was assigned a 100% value. The promoted enzyme production was $MgCl_2$, $K_2HPO_4$, $CaCl_2$ and $MnCl_2$, relative activity of accounting for 118.0%, 115%, 112%, 108% and 31.01%, respectively. Whereas $CoCl_2$, $FeCl_2$, $ZnCl_2$ and $CuCl_2$ each comprised only 86%, 45%, 45%,

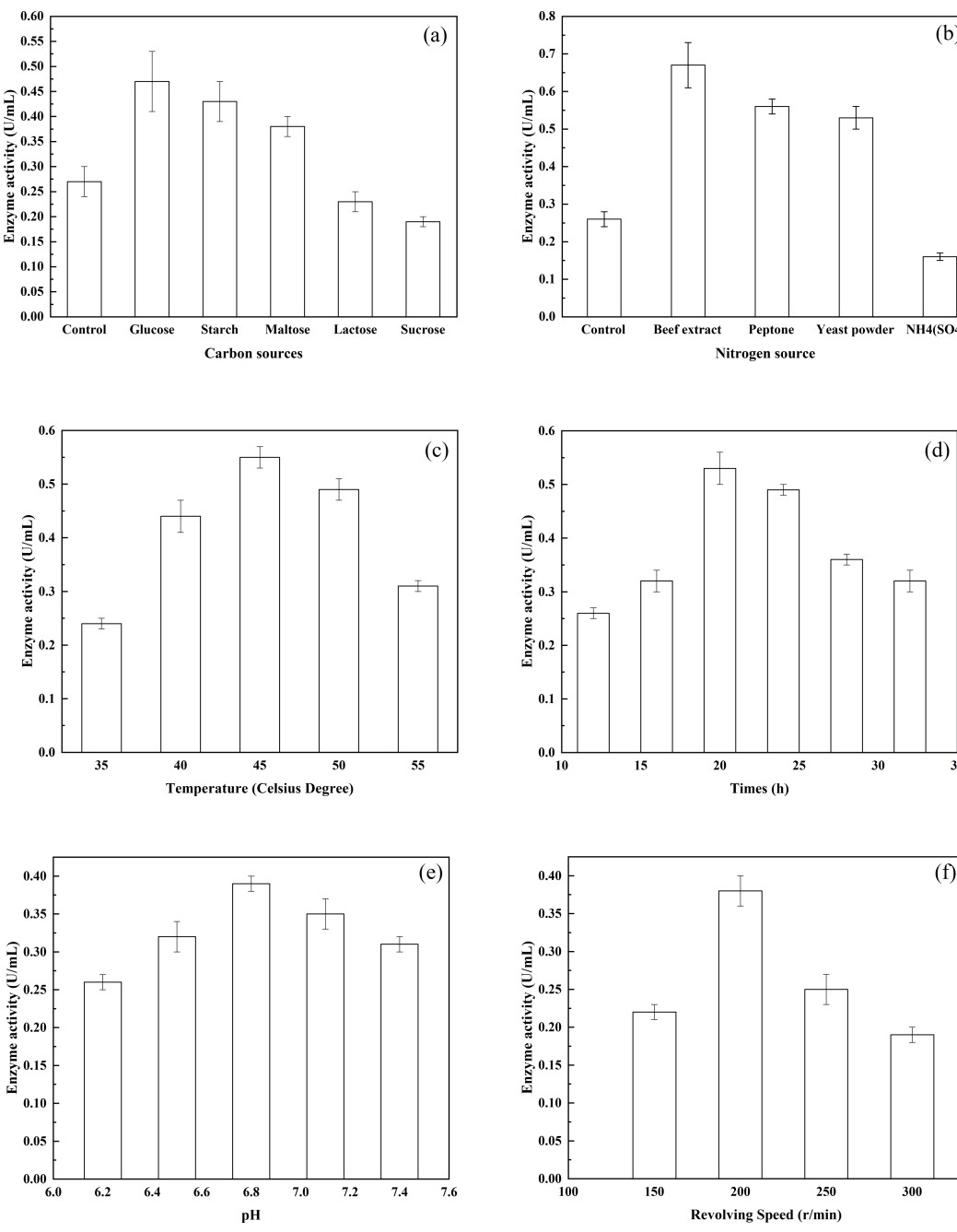

**Figure 2** **The effects of various environmental factors on bacterial enzyme production.** (A) The effects of various carbon sources on enzyme production; (B) the effects of various nitrogen sources on enzyme production; (C) the effects of different temperatures on enzyme production; (D) the effects of different pH on enzyme production; and (E) the effects of different rotational speeds on enzyme production.

8%, respectively. Based on the above experimental results, starch, peptone, beef extract, yeast powder, $MgCl_2$ and $K_2HPO_4$ were selected as the components of the optimized fermentation medium. To determine the effect of fermentation temperature on enzyme production by *G. stearothermophilus* U2, fermentation was carried out for 24 h, at pH 7.0,

a rotational speed of 200 r/min and at varying temperatures, and enzyme activities were measured. As shown in Fig. 2C, enzyme activity was very low at 30 °C but rather high at 42 °C. Therefore, the fermentation temperature was controlled at 37 °C. To determine the highest temperature of the enzyme, fermentation temperature continuing increased up to 55 °C, enzyme activity is kept half as much as 37 °C. The thermal stability of alpha-transglucosidase was further tested during the long incubation period. The enzyme was thermostable and no loss in activity was found below 37 °C, but it was inactivated at 65 °C or more and completely lost its activity at 80 °C. Thus, alpha-transglucosidase produced by *G. stearothermophilus* U2 possess heat-resistant properties, which is very important role as thermal stability to produce IMOs in food industry. To determine the effect of culture time on enzyme production by *G. stearothermophilus* U2, the activity of the enzyme produced was measured after culturing the bacteria for varying lengths of time (Fig. 2D). The results showed that enzyme activity was the highest when the culture time was 20 h. To determine the effect of the initial pH on enzyme production by *G. stearothermophilus* U2, fermentation was carried out for 20 h at 37 °C and different pH levels, and enzyme activity was measured (Fig. 2E). The results showed that the optimal initial pH value was 6.8. Finally, the effect of shaker rotational speed on enzyme production by *G. stearothermophilus* U2 was examined. Enzyme production is an oxygen-consuming fermentation process. During fermentation, the ventilation volume was adjusted by changing the shaker rotational speed. After 20 h of fermentation, enzyme activity was measured (Fig. 2F). It was found that low rotational speeds and small ventilation volumes were not conducive to bacterial growth. However, premature autolysis of bacteria would occur if the rotational speed was too high, which led to decreased biomass and reduced enzyme activity. Therefore, the optimal rotational speed was 200 r/min.

## Fermentation experiments using 5 L and 20 L fermenters

Fermentation experiments were carried out in 5 L fermenters based on test tube and shaker flask cultures. The medium used in the experiments was an optimized liquid fermentation medium. The fermentation conditions were also optimized: 10% inoculum, pH 6.8, 37 °C and 20 h of incubation. The bacteria cultured in 10 ml test-tube (0.71 U/mL) were 1.2 times higher than those of the unoptimized cultivation (0.60 U/mL). The bacteria cultured in 10 ml test-tube were inoculated into 250 ml shake-flask and placed on a shaker. After culture for 20 h at 37 °C with vigorous stirring (200 r/min), enzyme activity was evaluated. Subsequently, *G. stearothermophilus* that had been cultured for 20 h in 250 ml shake-flask were transferred into the 5 L fermentation tank. After 20 h of culture (rotational speed, 200 r/min; ventilation volume, 1.7; 37 °C), enzyme activity was measured. Whereas 250 ml shake flask cultivation and 5 L fermentation tank comprised 1.42 U/mL and 2.62 U/mL, respectively. The bacteria cultured in 5 L fermentation tank were 4.4 times higher than those of the unoptimized cultivation. At the completion of the 20 L-scale fermentation experiments, bacteria cultures were stored at 4 °C for future assays. The bacteria were centrifuged in batches (centrifugation conditions: 10,000 r/min for 20 min). The resulting pellets were collected with acetic acid-sodium acetate buffer (pH 6.0) and phosphate buffer (pH 6.8) and were stored at −18 °C. Enzyme production by ten batches of fermentation

reaction were observed. Enzyme activity of ten batches from 15.44 to 48.25 U/mL, with weight of bacterial cells from 2.0 to 4.0 g, which was harvested from 130.30 L of bacterial culture and 456. 05 g of weight of the pellet.

## Immobilization of alpha-glucosidase on magnetic chitosan microspheres

Determination of the ratio of chitosan to $Fe_3O_4$: Various chitosan/$Fe_3O_4$ ratios and the resulting states of the system were observed. With increasing $Fe_3O_4$, increasing amounts of free $Fe_3O_4$ appeared in the system. No free chitosan or $Fe_3O_4$ was detected in the system when the chitosan/$Fe_3O_4$ ratio was 1:1. Therefore, the ratio of chitosan to $Fe_3O_4$ was set to 1:1, which ensured the highest material efficiency. There is a certain limit to the amount of alpha-glucosidase that can be immobilized by the given chitosan gel. To immobilize the maximum amount of alpha-glucosidase without wasting enzyme, the effect of enzyme dosage on enzyme immobilization was investigated. The results are shown in Fig. 3A. When the enzyme dosage was relatively low, the activity of the immobilized enzyme rose rapidly as the enzyme dosage increased. After the enzyme dosage reached 3 mL, the magnitude of the increase in the activity of the immobilized enzyme was reduced significantly, and the enzyme activity recovery rate decreased sharply. Therefore, 3 mL appeared to be an appropriate enzyme dose. Under such condition, the activity of the immobilized enzyme was 282.18 mmol/g, and the enzyme activity recovery rate was 64.87%. The effect of adsorption time on enzyme immobilization was examined, and the results are shown in Fig. 3B. A certain concentration of $\alpha$-glucosidase was added to the chitosan gel. The activity of the immobilized enzyme reached the maximum when the adsorption time was 2 h. Therefore, 2 h of adsorption was appropriate for enzyme immobilization. The effects of crosslinking temperature on enzyme immobilization are shown in Fig. 3C. As the crosslinking temperature rose, the activity of the immobilized enzyme increased accordingly. The activity of the immobilized enzyme reached the maximum when the crosslinking temperature was 50 °C. However, enzyme activity started to decline when the crosslinking temperature was further increased. Therefore, crosslinking at 50 °C was appropriate for enzyme immobilization. The effects of glutaraldehyde concentration on enzyme immobilization are shown in Fig. 3D. Glutaraldehyde was added to the chitosan gel that had adsorbed a certain amount of alpha-glucosidase to induce crosslinking. The activity of the immobilized enzyme reached the maximum when the glutaraldehyde concentration was 4%. However, enzyme activity declined rapidly as the glutaraldehyde concentration was further increased. It is likely that the enzyme was inactivated when the glutaraldehyde concentration exceeded a certain threshold. Therefore, a glutaraldehyde concentration of 4% was appropriate for enzyme immobilization. The effects of crosslinking time on enzyme immobilization are shown in Fig. 3E. Under the present experimental conditions, the activity of the immobilized enzyme reached the maximum when the crosslinking time was 1.5 h. Enzyme activity was then decreased as the crosslinking time was further prolonged. The decrease in enzyme activity might be due to the denaturation of the enzyme induced by over-crosslinking of the enzyme with glutaraldehyde. Therefore, a crosslinking time of 1.5 h was appropriate for enzyme immobilization.

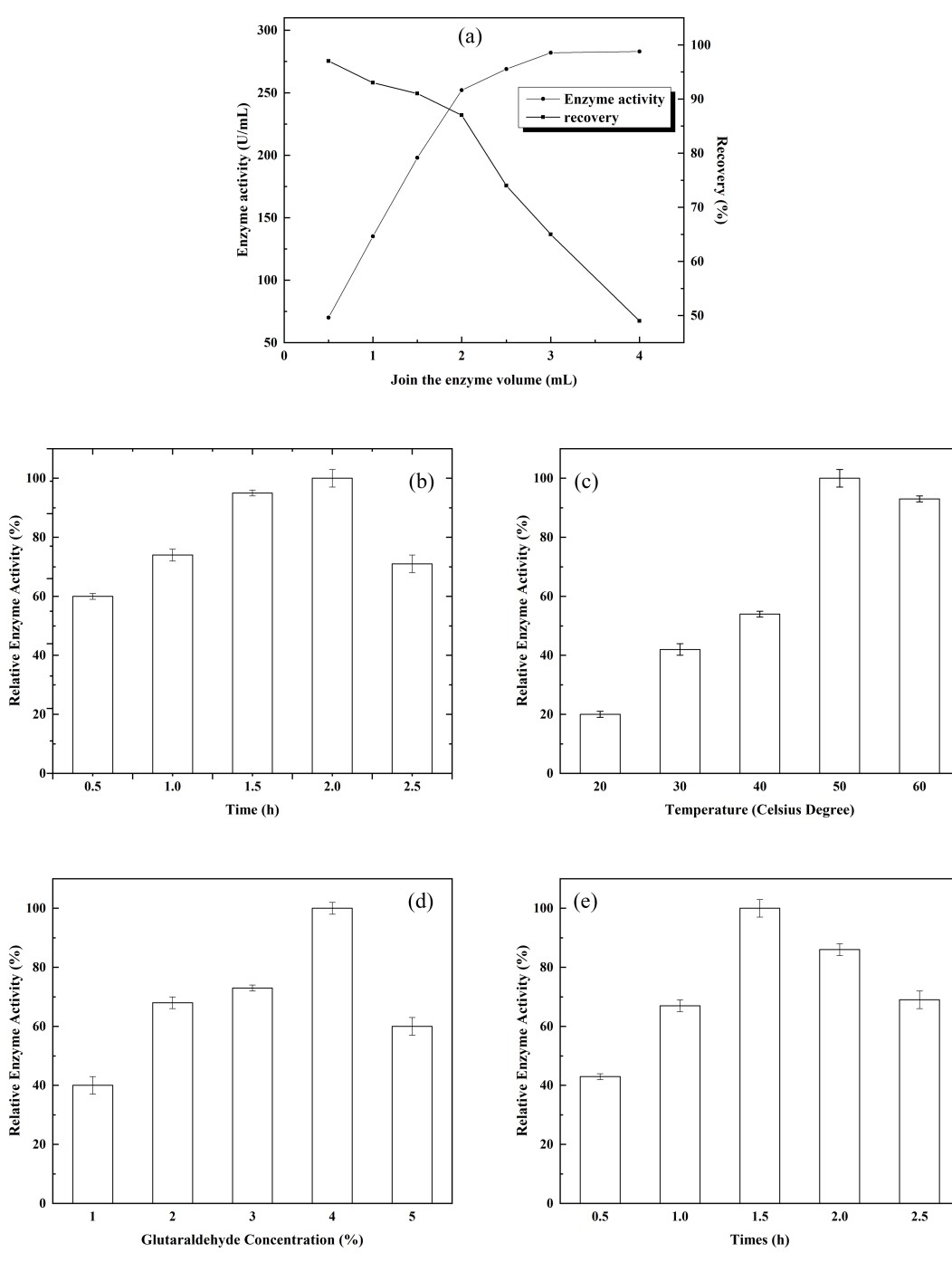

**Figure 3** **Immobilization of alpha-glucosidase on magnetic chitosan microspheres.** (A) The effect of enzyme dosage on enzyme immobilization; (B) the effects of adsorption time on enzyme immobilization; (C) the effect of crosslinking temperature on enzyme immobilization; (D) the effect of glutaraldehyde concentration on enzyme immobilization; and (E) the effect of crosslinking time on enzyme immobilization.

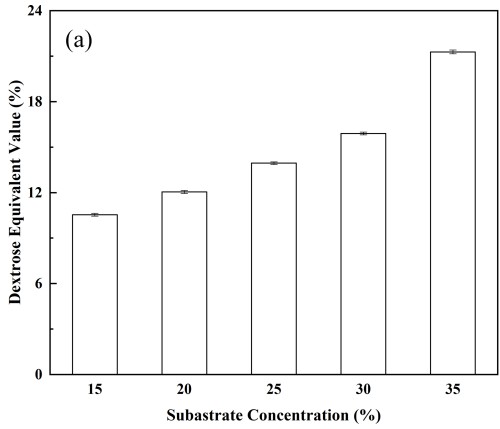

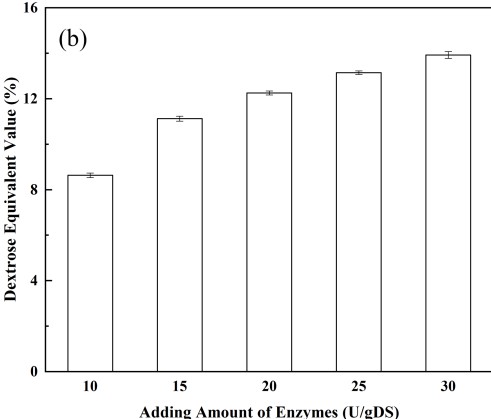
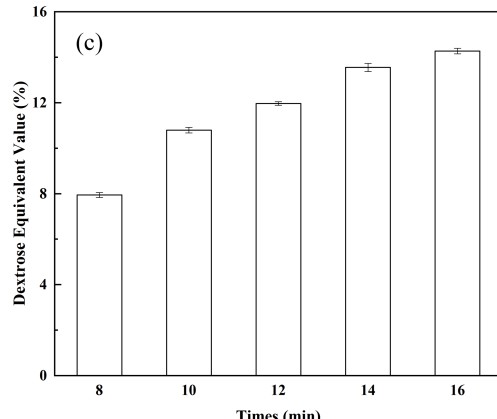

**Figure 4** **Optimization of the liquefaction process.** (A) The effect of substrate concentration on DE value; (B) the effect of enzyme dosage on DE value; and (C) the effect of liquefaction time on DE value.

## Preparation, detection and comparison of IMOs
### *Optimization of the liquefaction process*

The effect of substrate concentration on the DE value of the liquefied solution: substrate concentration is one of the key factors affecting the DE value of the liquefied solution. In addition, substrate concentration indirectly affects the degree of saccharification. As shown in Fig. 4A, the DE value of the liquefied solution rose as the corn starch concentration increased. When the corn starch concentration was 15%, the DE value of the liquefied solution was 10.54%. The DE value of the liquefied solution increased to 12.04% when the corn starch concentration increased to 20%. The corn starch slurry would become over-thickened if the concentration of corn starch was further increased, leading to incomplete liquefaction and the appearance of clot. Therefore, the corn starch slurry concentration should be controlled at approximately 20% when liquefying. The effect of enzyme dosage on the DE value of the liquefied solution: as shown in Fig. 4B, the DE value showed a constant upward trend as the dosage of thermostable alpha-amylase increased. The DE value of the liquefied solution reached 12.25% when the dosage of the liquefying enzyme

**Table 2  The results of orthogonal analysis of the liquefaction process.**

| Experiment number | A<br>Starch slurry concentration (%) | B<br>The dosage of liquefying enzyme (U/g) | C<br>Liquefaction time (min) | DE value (%) |
|---|---|---|---|---|
| 1 | 1(15) | 1(15) | 1(10) | 11.01 |
| 2 | 1 | 2(20) | 2(12) | 11.21 |
| 3 | 1 | 3(25) | 3(14) | 12.03 |
| 4 | 2(20) | 1 | 2 | 12.21 |
| 5 | 2 | 2 | 3 | 12.33 |
| 6 | 2 | 3 | 1 | 11.04 |
| 7 | 3(25) | 1 | 3 | 13.02 |
| 8 | 3 | 2 | 1 | 12.07 |
| 9 | 3 | 3 | 2 | 12.45 |

was 20 U/g. It may be seen that the highest conversion of corn starch (DE value 14.2%) was achieved with 30 U/g of adding amount of alpha-amylase. However, an incomplete liquefaction and the appearance of clot were obtained in more than 12.00% of the DE value of the liquefied solution. Obviously, the desired conversions could be achieved with lower enzyme concentrations. Therefore, indicating that the lower amount of alpha-amylase is sufficient for the effective conversion of the substrate. The effect of liquefaction time on the DE value of the liquefied solution: As shown in Fig. 4C, the DE value displayed a constant upward trend as the liquefaction time increased. The DE value of the liquefied solution reached 11.96% (a desired DE value for the liquefied solution) after 12 min of reaction. It may be seen from Fig. 4C that DE value of 14.6% is reached after 16 min, while a DE value of 11.96% could be obtained with 12 min of reaction. However, to our opinion, improving of the DE value is not economically justified because of the incomplete liquefaction. In addition, the longer exposure of the enzyme to high temperatures, which are needed for gelatinization of the corn starch granules and for achieving a good susceptibility to enzyme action, could lead to slight enzyme deactivation (*Mojović et al., 2006*). Therefore, it was concluded that the optimal liquefaction time was 12 min. Orthogonal tests of the liquefaction process have been investigated according to the design shown in Table 2. Corn starch was used as the raw material and subjected to liquefaction by thermostable alpha-amylase. After the alpha-amylase was inactivated, the DE value of the liquefied corn starch was determined. The DE value of the liquefied solution served as an index. The effects of corn starch slurry concentration, liquefying enzyme dosage and liquefaction time on the DE value were investigated under specific liquefaction conditions (pH 6.5 and water bath temperature of 95 °C). The results of the orthogonal test (shown in Table 2) were subjected to range analysis using SPSS 16.0 software. The DE value was used as an indicator. As shown in Table 3, the most important experimental factor influencing corn starch liquefaction was A, followed by C and then B (i.e., A > C > B). The optimal collocation of the factors was A2 > C1 > B1. Namely, the corn starch slurry concentration was 20%, the dosage of thermostable alpha-amylase was 15 U/g, and the liquefaction time was 10 min. Confirmatory testing was conducted based on the collocation of $A_2C_1B_1$, and

**Table 3 Range analysis of the DE value.**

| Serial number | A | B | C |
|---|---|---|---|
| K1 | 11.33 | 12.03 | 11.97 |
| K2 | 11.95 | 11.94 | 11.69 |
| K3 | 12.45 | 11.77 | 12.08 |
| R | 1.12 | 0.26 | 0.39 |

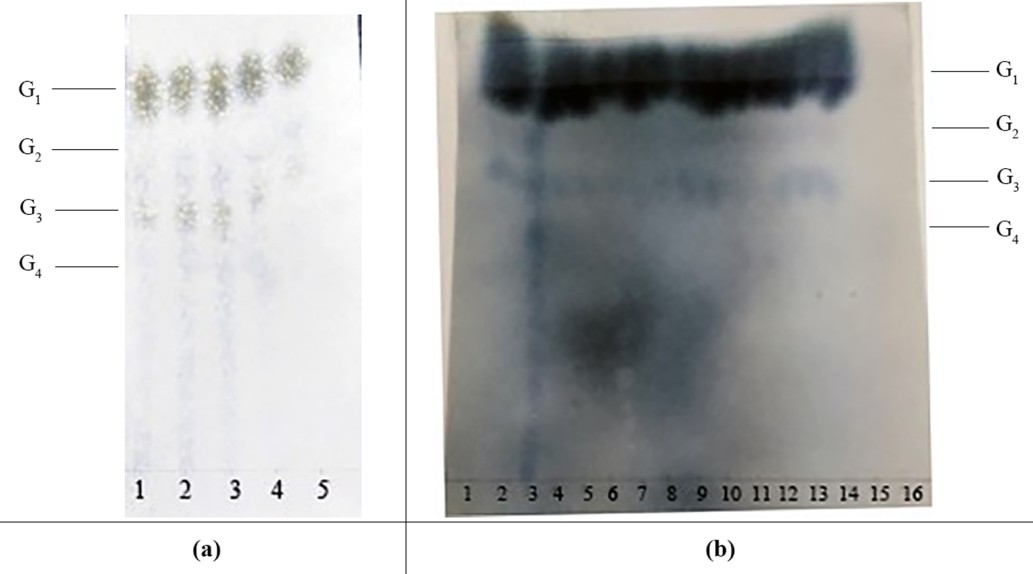

**Figure 5 The effects of various saccharification conditions on IMO content.** (A) Paper chromatography results showing the effect of glucoamylase dosage on IMO content. Lanes 1, 2, 3, 4, and 5 represent the different doses of glucoamylase used in the saccharification reaction (150 U/g, 200 U/g, 250 U/g, 300 U/g and 350 U/g, respectively). $G_1$, glucose; $G_2$, maltose; $G_3$, maltotriose; $G_4$, maltotetraose. (B) Paper chromatography results showing the effects of saccharification time and transglycosidation process on IMO content. $G_1$, glucose; $G_2$, maltose; $G_3$, maltotriose; $G_4$, maltotetraose. Lanes 1, 2, 3, 4 and 5 represent various saccharification times (3.0 h, 3.5 h, 4.0 h, 4.5 h and 5.0 h, respectively); lanes 6, 7, 8, 9 and 10 represent various doses of transglucosidase added to the reaction (0.25 U/g, 0.50 U/g, 0.75 U/g, 1.00 U/g and 1.25 U/g, respectively); lanes 11, 12, 13, 14 and 15 represent various transglycosidation times (38 h, 40 h, 42 h, 44 h and 48 h, respectively); lane 16 represents the control (distilled water).

the DE value of the liquefied solution was 12.05%. $A_1B_3C_3$-based confirmatory testing yielded a DE value of 12.45%. Intuitive analysis suggests that subsequent saccharification should not be pursued. Therefore, the optimal level of the factors was $A_2C_1B_1$.

### Optimization of the saccharification and transglucosidation processes

The effect of the dosage of glucoamylase on isomaltose content: IMOs were prepared using different doses of glucoamylase and then subjected to paper chromatography. The results are shown in Fig. 5A. The IMO content was highest when 250 U/g glucoamylase

**Table 4 Enzymatic activities of the transglucosidases from different sources and the differences between the IMOs produced using the transglucosidases.**

| Source of enzyme | Enzyme activity (U/mL) | Transglucosidation rate (%) |
| --- | --- | --- |
| This study | 3,856 ± 8.2 | 6.69 ± 0.2 |
| Amano Enzyme Inc., Japan | 6,015 ± 12.3 | 14.85 ± 1.2 |
| Shanghai Co., Ltd. | 5,616 ± 7.9 | 24.44 ± 3.1 |

**Table 5 Differences between the IMOs produced in the present study and commercially available IMOs.**

| Source of IMO | Glucose content (%) | Oligosaccharide content (%) |
| --- | --- | --- |
| This study | 9.81 | 41.74 |
| Shandong Co., Ltd. | 0.0 | 44.84 |
| Henan Co., Ltd. | 3.44 | 36.78 |

was used. The effect of saccharification time on isomaltose content: the experimental results are shown in Fig. 5B. The highest amount of IMO was produced after 4.5 h of saccharification. The effect of the dosage of alpha-transglucosidase on isomaltose content: the experimental results showed that the IMO content was highest when the dose of transglucosidase was 1.00 U/g. The effect of the duration of transglucosidation reaction on isomaltose content: the experimental results showed that the IMO content was highest when the duration of transglucosidation was 42 h. In the present study, the IMO syrups produced using 3 different transglucosidases were subjected to HPLC analysis. The HPLC results are shown in Figs. 6A–6C. The activities of the transglucosidases and the enzymatic conversion rates are shown in Table 4. The enzymatic activity of the transglucosidase made in this study was lower than the activity of the commercially available transglucosidases (the transglucosidases purchased from Amano Enzyme Inc., Japan and Shanghai Co., Ltd.). The transglucosidase made in our laboratory exhibited a transglucosidation rate of 6.69%, which was also lower compared with the transglucosidases purchased from Amano Enzyme Inc., Japan (14.85%) and Shanghai Co., Ltd (24.44%). Moreover, we found that glucose, maltose, isomaltose, maltotriose, panose and pentasaccharide were identified by the HPLC patterns, but except that of isomaltotriose and tetrasaccharide. Interestingly, maltotriose was mainly contains in IMO syrup that was produced using the transglucosidase purchased from Japan and Shanghai. These provide a reasonable explanation as to why the transglucosidation rate was lowest in our IMO syrup. In addition, the IMO prepared in our laboratory and the commercially available IMOs were analyzed by HPLC. The results are shown in Figs. 7A–7C. The contents of the oligosaccharides are shown in Table 5. The content of the oligosaccharides in the lab-made IMOs was 41.74%, which is lower compared with the IMO purchased from Shandong Co., Ltd. (44.84%), but higher than Henan Food-additive Co., Ltd (36.78%). Moreover, the glucose content was 9.81% for this study, 0.0% for Shandong Co., Ltd. and 3.44% for Henan Co., Ltd. On one hand, the results might be related to the fact that the saccharides were unpurified. On the other hand, How to decease glucose content and to increase content of the oligosaccharides will be an important core in the IMO industrial.

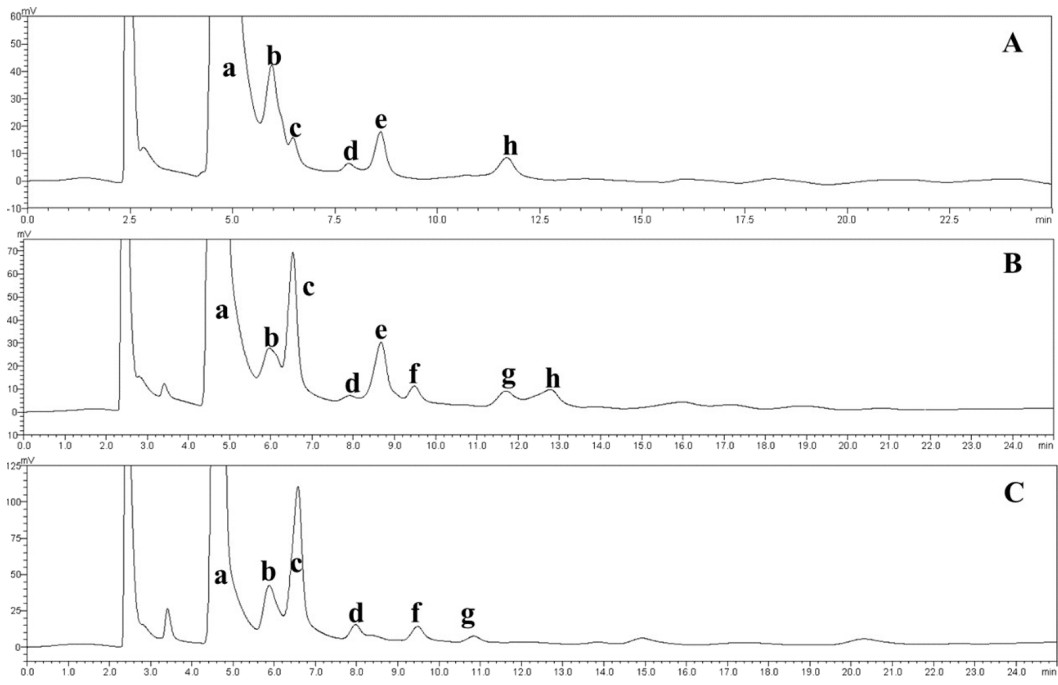

**Figure 6  HPLC chromatograms of the IMOs prepared using enzymes from different sources.** a, glucose; b, maltose; c, isomaltose; d, maltotriose; e, panose; f, isomaltotriose; g, tetrasaccharide; h, pentasaccharide. (A) Chromatogram of the IMO syrup that was produced using the transglucosidase prepared in our laboratory; (B) chromatogram of the IMO syrup that was produced using the transglucosidase purchased from Japan; (C) chromatogram of the IMO syrup that was produced using the transglucosidase purchased from Shanghai Co., Ltd., China.

## CONCLUSION

The optimized the conditions for biotransformation of starch into isomaltooligosaccharides using thermostable alpha-glucosidase from *G. stearothermophilus* U2 has been investigated. Under the optimal conditions, 5–20 L batch fermentation was explored in this work. The alpha-glucosidase activity was strongly inhibited by $Mn^{2+}$, $Co^{2+}$, $Fe^{2+}$, $Zn^{2+}$ and $Cu^{2+}$. IMOs were then prepared using chitosan membrane-immobilized alpha-glucosidase, beta-amylase, pullulanase, fungal alpha-amylase and starch as substrate. The mixed syrup that contained IMOs was evaluated and analyzed by thin-layer chromatography (TLC) and high-performance liquid chromatography (HPLC). In addition, small-scale preparation of IMOs was performed. The enzyme preparation was used in IMO production, resulting in an oligosaccharide content of 41.74% and the glucose content was 9.81%. These results are a strong indication that fundamental knowledge of alpha-transglucosidase-producing *G. stearothermophilus* as a potential application technique can be successfully used to prepare industrial IMOs.

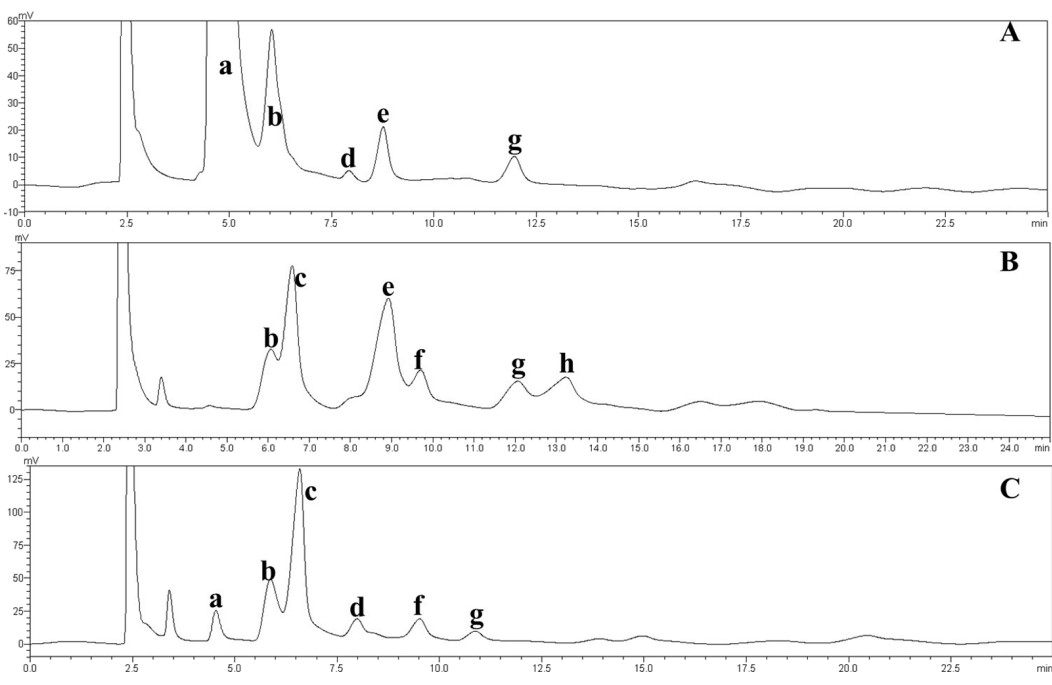

**Figure 7  HPLC chromatograms of the IMOs from different sources.** a, glucose; b, maltose; c, isomaltose; d, maltotriose; e, panose; f, isomaltotriose; g, tetrasaccharide; h, pentasaccharide. (A) Chromatogram of the IMO syrup prepared in this study; (B) chromatogram of the IMO produced by Shandong Co., Ltd.; (C) chromatogram of the IMO produced by Henan Co., Ltd., China.

## Funding

This work was supported by Technology Program of Gansu Province (Grant No. 1604FKCA110, Grant No. 1606RJZA087 and Grant No. 1610RJZA060), the Fundamental Research Funds for the Central Universities of China (Grant No. lzujbky-2017-197), the Project of Lanzhou City for Innovative and Entrepreneurial Talents (Grant No. 2017-RC-73) and Science and Technology Project of Lanzhou City (Grant No. 2015-3-93, Grant No. 2016-3-75 and Grant No. 2017-4-122). The funders had no role in study design, data collection and analysis, decision to publish, or preparation of the manuscript.

## Grant Disclosures

The following grant information was disclosed by the authors:
Technology Program of Gansu Province: 1604FKCA110, 1606RJZA087, 1610RJZA060.
Fundamental Research Funds for the Central Universities of China: lzujbky-2017-197.
Project of Lanzhou City for Innovative and Entrepreneurial Talents: 2017-RC-73.
Science and Technology Project of Lanzhou City: 2015-3-93, 2016-3-75, 2017-4-122.

## Competing Interests

The authors declare there are no competing interests.

## Author Contributions

- Peng Chen conceived and designed the experiments, performed the experiments, analyzed the data, contributed reagents/materials/analysis tools, prepared figures and/or tables, authored or reviewed drafts of the paper, approved the final draft.
- Ruixiang Xu and Jianhui Wang performed the experiments, analyzed the data, contributed reagents/materials/analysis tools, prepared figures and/or tables, authored or reviewed drafts of the paper.
- Zhengrong Wu and Lei Yan performed the experiments, prepared figures and/or tables, authored or reviewed drafts of the paper.
- Wenbin Zhao, Yuheng Liu and Wantong Ma performed the experiments.
- Xiaofeng Shi conceived and designed the experiments, analyzed the data, authored or reviewed drafts of the paper.
- Hongyu Li conceived and designed the experiments, analyzed the data, contributed reagents/materials/analysis tools, authored or reviewed drafts of the paper, approved the final draft.

## Data Availability

The raw measurements are provided in the Supplementary Files.

## Supplemental Information

Supplemental information for this article can be found online at http://dx.doi.org/10.7717/peerj.5086#supplemental-information.

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
