# Peer review of "Starch biotransformation into isomaltooligosaccharides using thermostable alpha-glucosidase from Geobacillus stearothermophilus"

_PeerJ, doi:10.7717/peerj.5086_

## Round 0.1 · original submission · Major Revisions

Thank you for your submission to PeerJ. I would like to to inform you that your manuscript has been recommended for publication subject to major revision in line with the listed referee reports. In addition to these referee reports, I also would like to suggest English improvements as some sentences are not very clear to understand. The statistics included within the figures should also be explained in details (i.e., how many times did you repeat the experiments; significance, etc...). Please have a consistency with the units given in the text; for example some places use "rpm" whereas the following text use "r/min". It was mentioned that the bacterial weight was calculated through absorbance measurements. This part needs more clarification as the calibration curve should also be included (or at least should be cited if you had previously given somewhere else).

Once again, thank you for submitting your manuscript to PeerJ and I look forward to receiving your revision.

Reviewer 1 ·

Basic reporting

The authors describe the application of a moderately thermostable Bacillus strain with alpha-glucosidase possessing transglycosidation activity. The manuscript shows the certain potential of the enzyme to be used in sugar industry for isomalto-oligosaccharides (IMOs). The data described in this paper provide important factors which affect the enzyme activity and production efficiency of IMOs.
However, the experimental settings, formats of data presentation and some methods in statistical analysis should be improved before publication to clarify the author’s statements.

Experimental design

The authors describe the application of a moderately thermostable bacillus strain with alpha-glucosidase possessing transglycosidation activity. The manuscript shows the certain potential of the enzyme to be used in sugar industry for isomalto-oligosaccharides (IMOs). The data described in this paper provide important factors which affect the enzyme activity and production efficiency of IMOs.

Some descriptions and experiments should be added as follows:

1. The origin of the strain used in this study should be stated in method section. For example, it was purchased from a XXX culture collection, isolated elsewhere (show references), or isolated in this study (show isolation method and origin of location and environment).

2. The thermal stability of the enzyme in this study and the other commercial enzymes should be compared. Because the authors referred to the thermal stability which is a bottleneck for this kind of enzymes in the introduction.

3. In Table 3, the description of experimental settings was not clear. More specified way should be employed to make the readers possible to conduct reproducible experiments.

Validity of the findings

The experimental settings, formats of data presentation and some methods in statistical analysis should be improved before publication to clarify the author’s statements.
Here are my opinions on specific points.

1. In Table 2, 7 9 and 10, The data should be showed with standard deviation from multiple experiments.

2. In Table 4 and 5, the data should be calculated in a statistical way and presented in text format in main manuscript, not in table format.

3. In Table 6, the description of the state of system is vague. The table should be removed, instead express in brief descriptions in the text.

4. In table 9, enzyme activity should be standardized though out a set of comparative study to demonstrate the efficiency of each enzyme.

5. In figure 5, please indicate the compound of the spot.

6. Experimental settings for enzyme evaluation in comparison with the commercial enzyme are not comprehensive. Please clearly state whether the experiments were conducted using immobilized enzymes or free liquid form enzymes.

Additional comments

The authors presented possible alternatives for commercial enzymes for IMOs. I think that the data presented here are significant to the reader for opening the way for further development of the same kind of the enzymes.
After appropriate revise according the suggestions, the paper can be published to promote the related research area.

Reviewer 2 ·

Basic reporting

The English is understandable though some grammatical mistakes should be rectified. For example, line 85, “and 5 L- and 20 L-scale fermentations were also carried out”; line 206, “previous”; delete: line 293, 303, 307, 313, 321 “The effect…immobilization”.
Fig. 3b, the figure abscissa description was pH? According to the context, it should be time.
Fig. 5: the isomaltose points should be clearly indicated on the figure.

Experimental design

The HPLC method should be described in detail. Line 211, What were the "minor modifications"?

Validity of the findings

Conclusion section, line 390-395, “The molecular mass…. The effects of pH and temperature on enzyme activity and stability…. The optimum temperature…65°C”. These results were not shown and discussed in the manuscript.

Additional comments

This manuscript reported the application of a thermostable alpha-glucosidase from Geobacillus stearotheriophilus for the transformation of starch into isomaltooligosaccharides. Generally, the techniques involved were useful for the industrial preparation of IMOs. However, the statements should be improved as follows:

1) The manuscript lacked comprehensive discussion of the results, especially the comparison with current IMOs production procedures in terms of thermostable alpha-glucosidase and immobilization of enzymes. The thermostable property of the investigated enzyme should be highlighted after some discussion and comparison with other known related enzymes.
2) Line 237, “However, glucose affected the yield of alpha-glucosidase.” The current results did not support this claim. It is recommended to add some data on the enzyme yield or cite some literatures if this is widely acknowledged.
3) Line 273, The results of fermentation experiments were in fact not described. The authors had only stated the fermentation process and conditions which should be described in the Materials and methods section. In addition, table 5, the enzyme activities in the 20 L-fermentation batches were not listed, described and discussed.
4) Line 342, what is the "comprehensive consideration"? The reason for the opt of enzyme dosage of 20 U/g and liquefaction time of 12 min should be stated more clearly.
5) Line 363, the HPLC results were not sufficiently described and discussed.
6) The purification of the thermostable alpha-glucosidase was not provided.
Other mistakes that need revision:
1) Line 22, 81, 406, “B. stearotheriophilus”, according to the microorganism used in the study , it should be Geobacillus stearotheriophilus (G. stearotheriophilus).
2) Line 24, “Chisomaltooligosaccharide” should be isomaltooligosaccharide.
3) Line 102, “p-NPG”, the full name should be given out in its first appearance.
4) Line 374, “Fig. 6(A), 5(B) and 5(C)”, I guess it should be “Fig. 6(A), (B) and (C)”.

---

## Round 0.2 · Minor Revisions

Thank you for your re-submission and your efforts to fulfill the referees’ comments. I would like to recommend minor revisions for the publication of your manuscript in PeerJ . I mainly suggest fulfilling the comments of our second referee on improving the conclusion and discussion sections further as well as the specific comments on the HPLC and optimization results (as stated in details below). I look forward to receiving the new improved version of your manuscript.

Reviewer 1 ·

Basic reporting

The manuscript have been revised properly according the reviewers comments.

Experimental design

The manuscript have been revised properly according the reviewers comments.

Validity of the findings

No comment

Additional comments

The manuscript have been revised properly according the reviewers comments. The paper will contribute to the progress of the related fields.

Reviewer 2 ·

Basic reporting

no comment

Experimental design

no comment

Validity of the findings

The Conclusion section only described what had been done in this study, while the main findings were almost missed.

Additional comments

1. The manuscript still lacked comprehensive discussion of the results, especially the comparison with current IMOs production procedures in terms of thermostable alpha-glucosidase and immobilization of enzymes. The thermostable property of the investigated enzyme should be highlighted after some discussion and comparison with other known related enzymes.
2. Section 3.4.1, the reason for the opt of enzyme dosage of 20 U/g and liquefaction time of 12 min should be stated more clearly. The authors did not make a substantive revision.
3. Section 3.4.2, the HPLC results were not sufficiently described and discussed, which actually was not revised by the authors according to the reviewer’s comments.

---

## Round 0.3 · accepted · Accept

Thank you for your re-submission. I am very pleased to inform you that your manuscript has been accepted for publication in PeerJ. Please also inform your co-authors of this decision. Thank you again for your submission to PeerJ. We look forward to seeing more of your work in the future.

#